# Evaluation of Quality of Life of Adult Patients with Celiac Disease in Argentina: From Questionnaire Validation to Assessment

**DOI:** 10.3390/ijerph17197051

**Published:** 2020-09-26

**Authors:** Nicole Selleski, Renata Puppin Zandonadi, Laura B. Milde, Lenora Gandolfi, Riccardo Pratesi, Winfred Häuser, Rosa Harumi Uenishi, Eduardo Yoshio Nakano, Claudia B. Pratesi

**Affiliations:** 1Interdisciplinary Laboratory of Biosciences and Celiac Disease Research Center, School of Medicine, University of Brasilia, Brasilia, DF 70910-900, Brazil; selleskinicole@gmail.com (N.S.); lenoragandolfi1@gmail.com (L.G.); pratesiunb@gmail.com (R.P.); rosa.uenishi@gmail.com (R.H.U.); 2Post-Graduate Program in Health Sciences, School of Health Sciences, University of Brasilia, Brasilia, DF 70910-900, Brazil; 3Department of Nutrition, School of Health Sciences, University of Brasilia, Brasilia, DF 70910-900, Brazil; renatapz@unb.br; 4Department of Food Science and Technology, School of Exact Sciences, Chemistry and Natural Sciences, University of Misiones, 3300 Posadas, Argentina; lauramilde@hotmail.com; 5Department of Internal Medicine I, Klinikum Saarbrücken, 66119 Saarbrücken, Germany; whaeuser@klinikum-saarbruecken.de; 6Department of Psychosomatic Medicine and Psychotherapy, Technische Universität München, 80333 München, Germany; 7Department of Statistics, University of Brasilia, Brasilia, DF 70910-900, Brazil; eynakano@gmail.com; 8College of Population Health, University of New Mexico, Albuquerque, NM 87131, USA

**Keywords:** quality of life, questionnaire, Argentina, celiac disease, validation

## Abstract

This cross-sectional study aimed to translate, culturally adapt, validate, and apply a Celiac Disease Quality of Life (CD-QoL) questionnaire to a representative sample of Argentina’s celiac population. A previously developed and validated questionnaire (Celiac Disease Questionnaire: CDQ) was chosen as a tool for assessing the health-related quality of Life (HRQoL) of adult celiac patients in Argentina. Therefore, the study was performed in four stages: (a) translation and re-translation of the CDQ to Argentinian-Spanish language; (b) cultural adaptation and semantic evaluation; based on the Delphi method (c) validation of the CDQ by applying it to a representative sample of Argentinian celiac patients; (d) statistical analysis of the data. The result of stages (a) and (b) was a translated and culturally adapted an Argentinian-Spanish version of the CDQ, which was generated after reaching consensus between the corresponding four (phase a) and 10 (phase b) professionals involved in the different phases of this process. Among them, we can cite bilingual healthcare professionals with extensive experience in research and celiac disease, celiac patients, gastroenterologists, general practitioners, dieticians, and psychologists. The resulting CDQ proved to be an appropriate measuring tool to assess the HRQoL of Argentinian celiac patients confirmed by a good fit in the confirmatory factor validity analysis (RMSEA < 0.001 and χ^2^ = 267.325, df = 313, *p* = 0.971) and high values of internal consistency (Cronbach’s alpha > 0.7). A total of 191 participants accessed the questionnaire, and 171 individuals from 20 out of 23 Argentinian states completed the questionnaire. There was no correlation between higher educational level nor marital status with QoL. Individuals on a strict gluten-free diet (GFD) and those who do not take antidepressants showed higher QoL. Male gender also presented better HRQoL. There was no correlation between differences in HRQoL and age of the respondent, age at diagnosis, symptoms at diagnosis, or having other chronic diseases. However, a significantly higher score of HRQoL was reported among those individuals who disclosed having knowledge of CD related national regulations and benefits. This study highlights the importance of maintaining current public health regulations that support chronic disease patients, such as celiac patients.

## 1. Introduction

Traditionally patients’ health has been measured from a biomedical perspective, and the broader impact on psychological and social factors has mostly been ignored. This broader impact is particularly significant in conditions such as celiac disease (CD), which encompass living with an autoimmune disease, and the challenge of managing a life-long gluten-free diet (GFD) [1,2,3]. These factors can impact patients’ subjective assessment of their general well-being Health-Related Quality of Life (HRQoL) and consequently influence the outcome of the disease prognosis and adherence to the treatment. HRQoL is a broad concept that can be defined as the patient’s subjective perception of the disease’s impact and the impact of the treatment(s) on the patients’ daily life, well-being, psychological health, and social functioning [4]. 

Celiac disease is defined as a systemic disorder characterized by a variable combination of signs, symptoms, and overproduction of specific antibodies; triggered by gluten intake in predisposed individuals [5]. Due to its global distribution and prevalence of about 1%, CD is considered a significant worldwide public health problem [6].

Currently, the only available treatment for CD is a strict GFD [1]. Patients that do not follow a GFD may face several complications. In addition to the difficulties imposed by the disease, they also face difficulties in following the GFD due to issues, such as lack of proper dietary guidance, financial difficulties, lack of health assistance and information, the long-rooted practice of consuming wheat products, and lack of cooking skills to prepare gluten-free meals [7], favoring treatment transgression. These treatment transgressions negatively impact the health and the QoL of celiac patients [2,8,9].

The perception of HRQoL in celiac patients has been stirring researchers’ interest as a measure to guide public policies and health professionals [2,10,11,12,13,14]. The burden of a lifelong GFD has demonstrated to be a significantly important factor in celiac patients’ lives. Shah et al. [15] demonstrated that this burden is higher than in many other chronic illnesses and even showed to be comparable to that of end-stage renal failure patients on dialysis [15]. This evidence supports the need to utilize a disease-specific instrument when evaluating HRQoL of celiac patients. Hauser et al. [16] developed a questionnaire to measure celiac patients’ perception of their HRQoL. This instrument was first applied to the German celiac population. The CD-QoL questionnaire (CDQ) evaluated four domains that affect CD patients; emotions, social, worries, and gastrointestinal symptoms [13,14,16,17,18]. The questionnaire was also cross-culturally validated and applied in France [17], Turkey [13], Italy [18], and Brazil [14]. Brazil was the only known Latin American country to apply this tool. To our knowledge, in Argentina, only two other studies evaluated the QoL of celiac adults [19,20]. Neither one used a specific CD questionnaire to evaluate the HRQoL of celiac adults in Argentina, as performed in other countries. These studies used the Short Form 36 Health Survey, the Gastrointestinal Symptoms Rating Scale, and the Beck Depression Inventory. Although very useful in assessing many gastrointestinal diseases patients’ QoL, these questionnaires alone lack CD-specific treatment-related questions.

Therefore, our study aimed to translate, culturally adapt, validate, and apply the CDQ to a representative sample of the celiac population in Argentina. We expect that the study data will allow future comparative research between different celiac populations in the world and help build new and update currently existing public policies for celiac patients´ care.

## 2. Materials and Methods 

This cross-sectional study was approved by the Universidad Nacional de Misiones – Facultad de Ciencias Exactas, Químicas y Naturales ethics committee, and followed the guidelines established by the Declaration of Helsinki. For the present study, the questionnaire (CDQ) developed and validated in 2007 by Häuser et al. [2] was chosen as a tool for assessing the HRQoL of adult celiac patients in Argentina. Unlike previously applied questionnaires, the CDQ looks at specific aspects of the disease, such as those associated with a gluten-free diet while still evaluating generic aspects of HRQoL. The combination of the generic and disease-specific questionnaire makes CDQ a unique tool that has already proven effective in assessing the QoL of celiac patients in other countries (Germany, Italy, France, Turkey, and Brazil) [2,12,13,14,21]. 

The study was performed in four stages: (a) translation and re-translation of the CDQ to Argentinian-Spanish language; (b) cultural adaptation and semantic evaluation; (c) validation of the CDQ by applying it to a representative sample of Argentinian celiac patients; (d) statistical analysis of the data.

### 2.1. Translation and Re-Translation of the CDQ to Argentinian-Spanish Language

The translation utilized the translation-back-translation methodology following the guidelines of Guillemin et al. [22]. At this stage, the questionnaire available in English was translated into Argentinian-Spanish by two bilingual health professionals with extensive experience in CD. The translations were carried out independently. After completing the translation, both translators, together with two other health professionals with expertise in the research and clinical areas of CD, met to discuss possible discrepancies between the translations, reached a consensus, and elaborated a corrected version of the questionnaire in Argentinian Spanish.

The consensus version was submitted for back translation into English by two independent bilingual translators. Finally, the translators involved in the translation-back-translation process met to verify that the generated version was compatible with the original questionnaire. The final version was submitted to the next step of cultural adaptation and semantic evaluation (Figure 1). 

### 2.2. Cultural Adaptation and Semantic Evaluation

The Delphi method was used for cultural adaptation and semantic evaluation of the version resulting from the previous step [23]. This method has been widely used in research involving semantic cross-cultural adaptation and subsequent validation of instruments due to its excellent reliability [24]. The Delphi method provides information on the precision with which the research tool measures what it intends to measure, allowing it to evaluate the instrument’s quality.

At this stage, the questionnaire was subjected to a cultural adaptation and semantic evaluation and content validation by a panel of ten Argentinian healthcare professionals (gastroenterologists, general practitioners, geneticists, dieticians, and psychologists) who have extensive experience in CD. These health professionals were different from those involved in the translation re-translation stage. In addition to health professionals, two celiac patients were also invited to participate. Therefore, the professionals represented the different possible areas involved in the diagnosis, treatment, and monitoring of CD, in addition to patients, allowing for a better perspective of the various dimensions involved.

Once the professionals gave their consent and agreed to participate in the study, they received a link to access the questionnaire on the SurveyMonkey^®^ (San Mateo, CA, USA) platform. The questionnaire contained the 28 translated questions into Argentinian-Spanish accompanied by their respective answer options. Each question provided an additional set of questions answered by the participants, which were scored on a 5-point Likert scale. After each answer, a space for comments and suggestions was available.

The questions were evaluated by their clarity, content validation, and semantic validation. The methodology used for calculating the agreement between experts was the Kendall coefficient of agreement (W). The coefficient values can vary between 0 and 1, with low values suggesting disagreement with the item. W-values ≥ 0.66 indicate the same standards of the evaluation were applied by experts [25]. For a question to be approved, there should be a minimum of 80% agreement between the experts (W-values ≥ 0.8) [14,26]. After consensus, the questionnaire was again evaluated by two bilingual translators to ensure that the changes made to adapt the questionnaire to the target population’s culture did not generate discrepancies between it and the English version (Figure 1).

### 2.3. Application of the CDQ in Argentina

The next stage of the research involved applying the Argentinian CDQ to the population of celiac patients over 18 years of age who lived in Argentina at the time of the survey. The survey was disseminated through sharing the hyperlink of the online version of the questionnaire uploaded into the SurveyMonkey^®^ platform. In addition to the Argentinian-CDQ 28 questions, 16 questions were added to assess socio-demographic data (final survey, Appendix A). The shared hyperlink allowed access to the platform through smartphones, tablets and or computers. The link was distributed through several social media networks to achieve the greatest possible coverage and representativeness of the Argentinian population. Additionally, informational posters containing both a hyperlink and a QR code to the online questionnaire were placed in stores that sell gluten-free products. The individual’s participation and or access to the questionnaire was only possible after reading the study description, inclusion, and exclusion criteria, and agree to the terms of the study. The inclusion criteria were: (i) celiac patient following GFD; (ii) lives in Argentina; (iii) ≥ 18 years old; (iv) agreed to participate in the study.

The different forms of distribution of the Argentinian CDQ, hosted in SurveyMonkey^®^ platform, allowed a nationwide availability of the Argentinian CDQ during July 2019 through February 2020.

### 2.4. Statistical Analysis 

According to Häuser [2] in group comparisons and different sex-ratios, multiple regression analysis scores were adjusted. If multiple measurements (such as in intervention studies) are performed, differences of ≥12 within the total score and ≥3 within each subscore can be regarded to be a minimum important clinical difference for intraindividual comparisons. For comparisons of groups, changes of ≥0.5 standard deviations can be regarded to be a minimum important clinical difference [27].

The statistical analysis followed the score proposed by previous studies that also applied the CDQ [2,14] in which a higher score indicates a higher perception of HRQoL. All questions were scored according to the participant´s answer. The subscale scores ranged between 7–49. The total score ranged between 28–196. If a question was left blank, a score matching the median value of the corresponding dimension was assigned to that question. The total score for each demographic and clinical dimension was calculated separately. When ≥25% of the questions were left blank by a respondent, the entire questionnaire was excluded from the analysis to prevent bias.

Descriptive statistical analysis was performed on results, and corresponding data (mean, median, standard deviation, floor effect, and ceiling effect) was presented by subscales (emotions, social, worries, and gastrointestinal) of the CDQ. We used Student’s *t*-test to compare the values of the subscales of CDQ with the socio-demographic variables. To evaluate the reliability within subscales (internal consistency), we used Chronbach alfa. To demonstrate the correlation between items within a scale, levels of Chronbach alfa should be ≥0.7 [28]. Factor/construct validity was assessed using Confirmatory Factor Analysis. For the evaluation of factor validity, two different statistical tests were applied; Root Mean Square Error of Approximation (RMSEA) (ranging from 0 to 1) and the Chi-squared test of minimum discrepancy [13]. The smaller values of RMSEA indicate a better model fit, considered acceptable when values were equal or less than 0.06 [19]. Data were processed using IBM SPSS (Statistical Package for Social Sciences) version 22 (IBM, Armonk, NY USA) and IBM SPSS AMOS (Analysis of Moment Structures) version 22 (IBM, Armonk, NY, USA). All applied tests considered a two-tailed hypothesis and a significance level of 5%. 

## 3. Results

### 3.1. Translation and Re-Translation; Cultural Adaptation and Semantic Evaluation

The English version of the CDQ was successfully translated following the process summarized in Figure 1 and detailed in item 2.1. Translators and health professionals reached consensus after the first round of discussion. The Spanish version resulting from that stage was submitted for cultural adaptation and semantic evaluation, this process took three rounds of evaluation. Round 1: In the first round of evaluation from this stage, two questions were only partially approved (approved with comments/suggestions of changes) by three different judges. Changes were made to the corresponding questions and they were resubmitted for judges evaluation. Round 2: In this second round of evaluation only one of the previous questions remained partially approved by two of the judges. Round 3: judges agreed to the changes made and the survey tool approved by judges was submitted for ethics committee approval. After the ethics committee revision and approval of the final version, the questionnaire was made available and distributed for application in the Argentinian celiac population.

### 3.2. Internal Consistency and Construct Validity of the Argentinian CDQ

The internal consistency was verified by the Cronbach’s alpha showing concordance between answers of all four domains of the CDQ (Cronbach’s alpha > 0.7) [29]. Ceiling and Floor effect range from 0% to 6.5%. Detailed data from descriptive statistical analysis (mean, median, standard deviation, floor effect, and ceiling effect) and internal consistency can be found in Table 1.

Confirmatory factor analysis was used to examine the factor/construct validity. The four domains (emotion, social, worries, and gastrointestinal) had a good fit in the confirmatory factor analysis (RMSEA < 0.001 and χ^2^ = 267.325, df = 313, *p* = 0.971).

### 3.3. Application of the CDQ in Argentina 

A total of 191 participants accessed the questionnaire; 171 of them completed the questionnaire. The twenty other questionnaires were excluded due to being either not (at least 75%) filled out or not at all. Among the 171 completed questionnaires, 149 were from female participants (88.7%)-four participant did not disclose their gender-, age between 18 and >60 years old, and belonging to 20 (out of 23) different Argentinian states. The questionnaire took, on average, eight minutes to complete. The characteristics of respondents of the CDQ and subcategories are presented in Table 2.

Males showed significantly higher scores of total HRQoL than females (145.19 vs. 121.73; *p* = 0.006). The results for all domains were also higher for males than for females; however, only statistically significant in the emotional and gastrointestinal (*p* < 0.001 and 0.001, respectively).

The GFD category was divided into two possible outcomes the respondents that answered “always” were considered to be on a GFD (indicated as yes in Table 2); those who answered, “usually,” “sometimes,” “hardly ever,” “never” were not considered to be on a GFD. Participants on a strict GFD showed higher scores of HRQoL in all evaluated domains, but those results were only statistically significant for gastrointestinal and total (*p* < 0.001 and 0.029, respectively).

The marital status category was divided into, stable relationship (married or with a live-in partner) or not in a stable relationship (single, divorced, or widowed). The marital status did not influence the total HRQoL nor in any domain of QoL (*p* > 0.05). Also, there was no correlation between the age of the respondent, age at diagnosis, symptoms at diagnosis, higher educational levels, or having other chronic diseases and differences in general HRQoL or in any of the domains (Table 2).

Participants who were not taking antidepressants at the time of the survey disclosed higher levels of HRQoL than those taking antidepressants in the emotional domain (26.39 vs. 21.27; *p* < 0.028). Other domains showed variable results with no statistical significance.

Regarding the subcategories related to celiac legislation, results show that those participants who were aware of the benefits presented significantly higher scores of total HRQoL. No statistically significant differences in HRQoL were evidenced between those who used and did not use the benefits stated in the celiac legislation except for the social domain (39.39 vs. 34.89; *p* < 0.012) (Table 2).

## 4. Discussion

The World Health Organization (WHO) defines health as “a state of complete physical, mental, and social well-being” [30]. Patients with autoimmune diseases, such as CD, depends on lifelong dietary restrictions imposed by the treatment. The impact on the patient’s HRQoL on those cases may be related not only to the disease itself but also to the recommended treatment (i.e., a gluten-free diet). Shah et al. [15] demonstrated that the burden of a life-long GFD could be considerably higher for many celiac patients than those affected by other chronic illnesses and similar to those of end-stage renal failure patients on dialysis. In such cases, QoL generic measuring tools may not be sufficient to assess the variables affecting the patient’s HRQoL [31,32].

In this study, a disease specific HRQoL measuring tool previously used in other European and South American countries [2,13,14,16,18] was translated, culturally adapted, and validated to be used in the Argentinian celiac population. A hyperlink and QR code were used to distribute the online version of the questionnaire. This allowed for a wider distribution of the measuring tool resulting in a more representative sample as well as assuring the anonymity of the participants. Anonymous answers reduce the bias associated with the discomfort or shame to report transgressions from treatment and therefore allow a more accurate correlation between the real status of GFD compliance.

The final instrument proved to be a reliable tool to the assessment of HRQoL in the Argentinian celiac population confirmed by a good fit in the confirmatory factor validity analysis (RMSEA < 0.001 and χ^2^ = 267.325, df = 313, *p* = 0.971) and high values of internal consistency (Cronbach’s alpha > 0.7) [29]. Ceiling and Floor effect did not reach concerning values (0 to 6.5%), meaning the final instrument can be used to assess HRQoL with precision in the studied population. Even the subscale showed the highest score of ceiling effect −6.5%; social- was more than two to almost five times lower than those reported in previous studies applying the same instrument (between 15.9 and 32.0%). The instrument was widely distributed along with the country as expected (results came from 20 out of 23 states), and its content had a good acceptance from the participants demonstrated by low levels of unanswered questions. These questions were most frequently related to financial income and gender, which could be sensitive issues for some individuals, as demonstrated in other previously performed studies [2,17,18].

In agreement with previously performed studies, females present lower HRQoL than males, in the present study, females’ scores for the CDQ were significantly lower than those in the emotional and gastrointestinal domains as the general HRQoL [14,33]. Some studies showed that lower QoL among women is mainly associated with the distress caused by daily life restrictions and perceive a higher burden than expressed by males [34,35,36].

Individuals on a strict GFD showed higher general QoL which was consistent with previous studies performed in a sample of Argentinian celiac patients and when applying the same QoL measuring tool to the Brazilian population [14,20]. Additionally, individuals on a strict GFD showed higher QoL regarding the gastrointestinal subscale than those that do not follow a strict GFD. These differences are because individuals with CD, that do not follow a strict GFD, generally suffer from several gastrointestinal manifestations resulting in reduced QoL [37,38,39]. 

The fact that individuals on a strict GFD showed significantly higher levels of QoL than those that were not (*p* = 0.029) suggests that, like celiac patients from other countries [2,14], following strict GFD results in improved general well-being (health-related quality of life). A significantly higher HRQoL was found among those patients who were aware of the existence of benefits related to the celiac legislation (*p* < 0.001 and *p* = 0.004, respectively), regardless of whether or not they made use of those benefits (*p* = 0.111). This improved general well-being due to the knowledge of benefits raised an interesting point that could be associated with patients feeling better understood and emotionally supported. In addition to covering clinical and food-related aspects of the disease, the legislation also encompasses broader actions that promote education on CD, and classes on the preparation of gluten-free meals in schools, restaurants, and community centers, resulting in greater social inclusion for those with CD. 

Although it was not our objective to evaluate the adherence to GFD, we found a high proportion of respondents answered they were “always” on a GFD, 83.9%. This finding is in concordance with the results obtained by Nachman et al. [20], who performed studies in the Argentinian celiac population and also with others like Cabrera-Chávez et al. [40] who assessed the adherence to GFD in the general population from Santa Fé Argentina. We believe this could be associated with the fact that Argentina, unlike other South American countries, has regulations that protect celiac patients by promoting public health actions concerning the training of healthcare professionals in different aspects of the disease (diagnoses, treatment, and follow-up). The Argentinian Public Health Department monitors the dissemination of information related to CD and also establishes the obligation of availability of care related to the diagnoses, follow-up, and treatment, among others. Since the only treatment for CD is a lifelong GFD, as mentioned before, the monitoring also regulates the availability of gluten-free products, which should be at least partially guaranteed by the government or private/public healthcare provider. The quality of industrially packaged gluten-free food, meals, non-packaged food served in restaurants, schools, and other places offering food for their employees, clients, and users are also monitored [41,42]. The most recent update of their CD law [42] also specifies the importance of patient training in preparing gluten-free meals and the regulation and monitoring of gluten content in medicines. 

Unlike Brazil, the only other South American country where this questionnaire was applied [14], there was no difference in HRQoL among those with different educational level or marital status. There was also no correlation between differences in HRQoL and age of the respondent, age at diagnosis, symptoms at diagnosis, or having other chronic diseases.

Patients who reported not using antidepressants also showed a significantly higher QoL. Interestingly, when analyzing the proportion of celiac patients that use antidepressants, we also noticed a considerably lower number than those displayed in a recent study in Brazil (7% vs. 17%). This may be because Argentinians perceive therapy as an essential part of personal development and health. Argentina has the largest number of phycologists per capita, and the highest number of citizens in therapy [43]. The availability and high frequency of therapists, along with the CD regulations, could be the reason why Argentinian celiac patients show these high values of HRQoL when compared with closely related countries like Brazil.

### Potential Limitations

A possible limitation of this study may be that our sample had a higher proportion of female respondents (88.7%). However, other studies with gluten-related disorders from other countries also exhibited this trend, with over 70% of female respondents [2,14,32,33]. Previous studies on HRQoL of celiac patients performed in the Argentinian population showed as high as 90% of female respondents [20]. Moreover, females tend to generally be more concerned about their health and, therefore, participate more in health surveys than males [14,44]. Another potential limitation could be the selection bias due to the online distribution of the survey. However, according to a recent international survey, 93.1% of the Argentinians have internet access, and many cities provide free internet access in public places [45,46].

Since the study’s main goal was to validate a newly adapted measuring tool, the period in which the questionnaire was available to be accessed was only seven months. Many categories and subcategories showed differences that did not reach the significance established level. A more extended period of availability and broader dissemination of the questionnaire could overcome this limitation by increasing the number of respondents.

## 5. Conclusions

The resulting CDQ proved to be an appropriate measuring tool to assess the HRQoL of Argentinian celiac patients. Male gender and those on a strict GFD presented better HRQoL. Unlike Brazil, the only other South American country where this questionnaire was applied, there was no difference in HRQoL among those with different educational level or marital status. There was also no correlation between differences in HRQoL and age of the respondent, age at diagnosis, symptoms at diagnosis, or having other chronic diseases. However, a significantly higher score of HRQoL was reported among those individuals who disclosed being aware and knowing the benefits of the CD-related national regulations.

We hope our study highlighted the importance of having and maintaining current public health regulations that support chronic disease patients like celiac individuals. We hope that our results inspire further studies in the area of HRQoL, for Argentina and other countries. We believe comparative studies from other regions may promote actions and shared knowledge on improving patients’ well-being.

## Figures and Tables

**Figure 1 ijerph-17-07051-f001:**
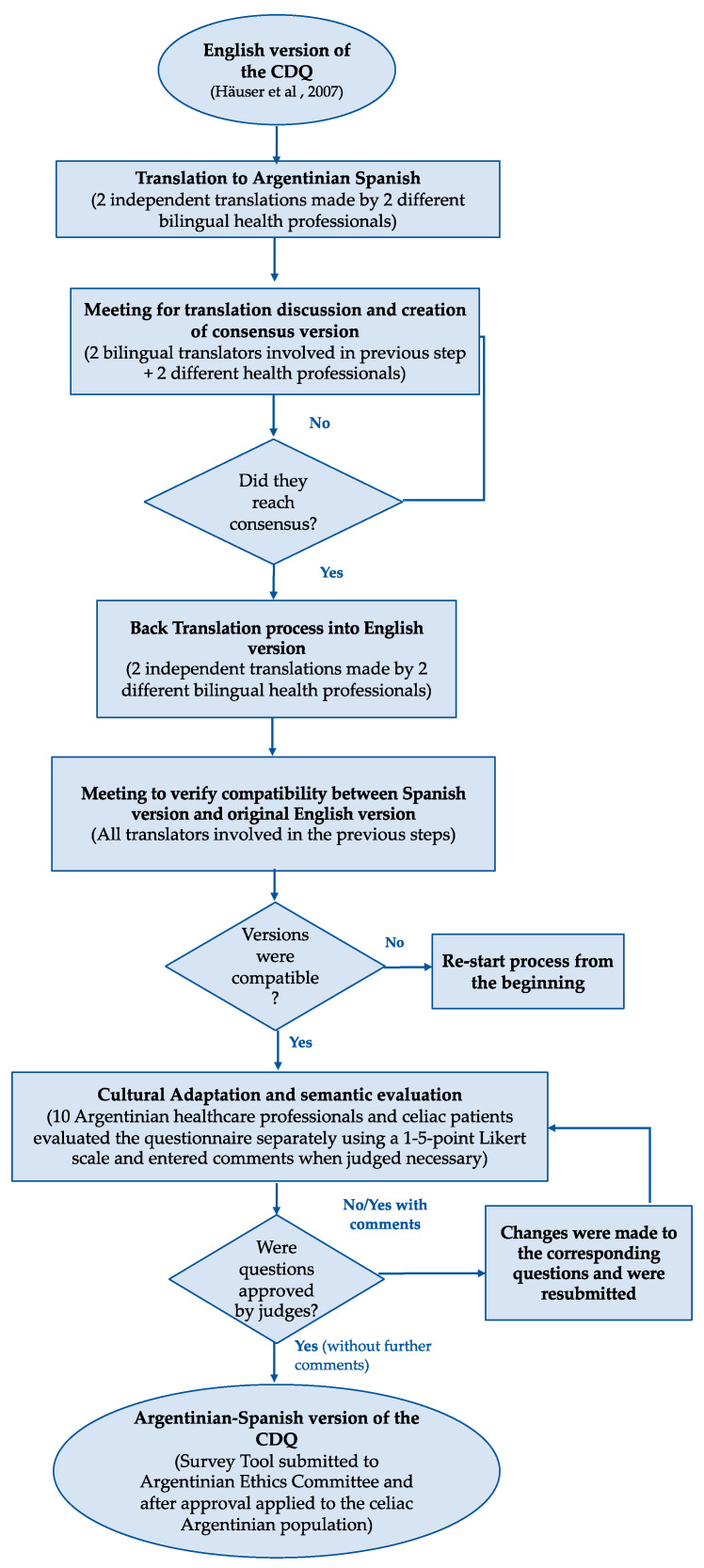
Flowchart of stages of the Argentinian-Spanish version of celiac disease questionnaire (CDQ).

**Table 1 ijerph-17-07051-t001:** Description of internal consistency of the Argentinian CDQ.

	Mean (SD)	Median (IQR)	Range	Floor Effect (%)	Ceiling Effect (%)	Internal Consistency (Alpha Cronbach)
Emotional	26.07 (10.38)	24 (18–34)	7–49	0%	1%	0.922
Social	35.8 (9.25)	37 (28–44)	10–49	0%	6%	0.832
Worries	28.82 (10.11)	28 (31–37)	7–49	0%	1%	0.771
Gastrointestinal	33.77 (9.24)	34 (27–41)	9–49	0%	3%	0.849
Total Score	124.14 (32.44)	123 (99–149)	34–96	0%	0%	0.934

SD—Standard deviation; IQR—Interquartile range. The total scale and its subtotals were consistent (alpha > 0.7).

**Table 2 ijerph-17-07051-t002:** Sub-scores of the CD-QoL scale subcategorized by the characteristics of participants.

	Emotion		Social		Worries		Gastrointestinal		Total	
	Mean (SD)	*p*	Mean (SD)	*p*	Mean (SD)	*p*	Mean (SD)	*p*	Mean (SD)	*p*
Gender *										
Female (*n* = 149)	25.12 (10.16)	<0.001	35.26 (3.38)	0.079	28.48 (10.16)	0.298	33.02 (9.32)	0.001	121.73 (32.44)	0.006
Male (*n* = 18)	34.11 (9.49)		39.33 (7.84)		31.25 (9.38)		40.56 (5.36)		145.19 (26.45)	
Present Age *										
≤39 years (*n* = 112)	25.67 (10.47)	0.423	35.45 (9.12)	0.568	28.24 (10.22)	0.293	33.05 (9.34)	0.134	121.82 (32.11)	0.197
≥40 years (*n* = 58)	27.02 (10.23)		36.31 (9.53)		29.98 (9.97)		35.29 (8.95)		128.70 (33.12)	
Age at diagnosis *										
≤30 years (*n* = 93)	27.15 (10.84)	0.103	36.06 (8.91)	0.532	29.03 (10.67)	0.750	33.51 (9.92)	0.698	124.90 (33.96)	0.645
≥31 years (*n* = 72)	24.49 (9.68)		35.14 (9.58)		28.54 (9.17)		34.06 (8.05)		122.51 (30.44)	
Symptoms at diagnosis *										
No (*n* = 40)	24.02 (8.53)	0.109	35.15 (9.34)	0.607	28.22 (10.35)	0.674	34.68 (7.60)	0.480	122.08 (28.45)	0.646
Yes (*n* = 130)	26.70 (10.84)		36.02 (9.26)		29.00 (10.07)		33.49 (9.69)		124.79 (33.69)	
College education *										
No (*n* = 83)	25.01 (10.83)	0.228	34.99 (9.73)	0.352	27.61 (10.35)	0.124	32.57 (9.22)	0.093	119.85 (33.40)	0.118
Yes (*n* = 82)	26.98 (9.91)		36.34 (8.61)		30.03 (9.57)		34.96 (8.92)		127.86 (31.04)	
Marital status *^,^**°**										
Stable relationship (*n* = 63)	25.02 (10.84)	0.395	34.97 (9.71)	0.478	28.95 (10.38)	0.894	33.33 (9.20)	0.647	121.63 (34.29)	0.516
Not in a stable relationship (*n* = 101)	26.44 (10.07)		36.03 (8.91)		28.74 (9.87)		34.01 (9.16)		125.07 (31.33)	
Gluten-free diet *^, ±^										
No (*n* = 27)	22.74 (9.32)	0.067	33.59 (9.39)	0.173	27.30 (10.20)	0.352	28.33 (9.66)	<0.001	111.96 (32.71)	0.029
Yes (*n* = 141)	26.75 (10.53)		36.26 (9.23)		29.27 (10.05)		34.96 (8.71)		126.91 (32.03)	
Celiac legislation: Aware of Legislation *										
No (*n* = 41)	21.46 (8.58)	<0.001	30.49 (7.94)	<0.001	23.70 (9.97)	<0.001	29.68 (8.82)	0.001	103.87 (25.63)	<0.001
Yes (*n* = 128)	27.53 (10.54)		37.39 (9.07)		30.57 (9.53)		35.23 (8.87)		130.43 (31.86)	
Celiac legislation: Aware of benefits *										
No (*n* = 111)	24.51 (9.85)	0.007	34.71 (9.13)	0.040	27.62 (10.13)	0.021	32.79 (9.26)	0.032	118.98 (30.88)	0.004
Yes (*n* = 58)	29.02 (10.88)		37.79 (9.29)		31.38 (9.49)		35.97 (8.62)		134.16 (33.26)	
Celiac legislation: uses of benefit? *										
No (*n* = 136)	25.67 (10.16)	0.324	34.89 (9.39)	0.012	28.46 (10.26)	0.222	33.46 (9.43)	0.220	122.34 (32.94)	0.111
Yes (*n* = 33)	27.67 (11.40)		39.39 (7.92)		30.88 (9.00)		35.64 (7.76)		132.53 (29.46)	
Other chronic diseases *										
No (*n* = 98)	26.78 (10.80)	0.303	36.28 (8.64)	0.444	27.67 (9.52)	0.086	34.05 (8.98)	0.464	123.97 (31.07)	0.937
Yes (*n* = 72)	25.11 (9.76)		35.1 (10.06)		30.38 (10.74)		33.39 (9.63)		124.37 (34.47)	
Use of antidepressants *										
No (*n* = 158)	26.39 (10.56)	0.028	36.01 (9.36)	0.361	28.71 (10.27)	0.622	33.91 (9.33)	0.470	124.66 (32.97)	0.435
Yes (*n* = 12)	21.27 (6.33)		33.36 (8.00)		30.27 (7.81)		31.82 (8.38)		116.73 (23.68)	

* Student´s t-test. ° Marital status: Stable relationship (married or with a live-in partner); Not in a stable relationship (single, divorced, or widowed)). ^±^ Gluten-free diet adherence: Yes (always); No (usually, sometimes, hardly ever, never).

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
