# Peer review of "Evaluation of Quality of Life of Adult Patients with Celiac Disease in Argentina: From Questionnaire Validation to Assessment"

_ijerph, 2020, doi:10.3390/ijerph17197051_

Round 1
Reviewer 1 Report
This is an interesting study analyzing the characteristics of a new HRQL questionnaire in Argentina, applied to adults with celiac disease
The study is interesting and well designed and performed
I have included some recommendations and some queries in order to clarify some points and improve jis quality, in general
Reviewer´s evaluation :
Q1. Title :
I would suggest to include “Evaluation of” before Quality of Life
Q2. : Authors :
I would like to know the reasons to include several nacionalities like Brasil and Germany among the list of authors
Q3.: Abstract :
Is OK
Q4 : Introduction :
Is OK
Q5 : Materials and Methods :
Everything is correct
Q6 : Results :
Table 1, It is very informative and clear
Table 2, shows the Sub-scores of the CD-QoL
Which is the reason of a greater preponderance of Female=149 over Male =18, this makes a proportion of 8.27/1. In the CD the relaionship between females and Males is around 2/1.?
Do not consider this great participation of females into the study may have a n important difference in all the results of the study observed?
Q7 : Discussion
Line 291 : The phrase “The most recent update of the CD law is repeated”, please eliminate one of them
Q8 : Conclusions
You can remark that the study was not presenctial and this way of action may have influenced also n the results obtained
Instead of asking about the influence of vaious factors in the last 2 weeks woiuld be better to as aboute the last month
Q9 : References are good and well selected
Author Response
Reviewer 1
Thank you for your comments, questions and for taking your time to review our study.
Q1. Title:
I would suggest including “Evaluation of” before Quality of Life
A1. Title has been modified to include reviewer’s suggestion.
Q2.: Authors:
I would like to know the reasons to include several nationalities like Brazil and Germany among the list of authors.
A2: The original questionnaire was developed by a German research group led by W.H., he helped during the conceptualization phase of this work and the final review and editing from the manuscript. N.S. is a PhD student from a Brazilian university however, she is from Argentina. She also did her Bachelor´s degree in Argentina where she worked with L.B.M. in a celiac disease volunteer teaching program. Both research groups from Argentina (L.B.M) and Brazil (C.B.P.; R.H.U.; L.G.; R.P.; N.S.; R.P.Z. and E.Y.N.) worked closely together to develop this study. Although C.B.P. is currently affiliated to a US institution, she was affiliated to Universidad de Brasilia at the time this study started, and she continued working closely together with the previously mentioned Brazilian research group where she did her PhD before.
Abstract:
Is OK
Introduction:
Is OK
Materials and Methods:
Everything is correct
Results:
Table 1, It is very informative and clear
Table 2, shows the Sub-scores of the CD-QoL
Q3: Which is the reason of a greater preponderance of Female=149 over Male =18, this makes a proportion of 8.27/1. In the CD the relationship between females and Males is around 2/1.?
Do not consider this great participation of females into the study may have a n important difference in all the results of the study observed?
A3: Although we can´t confirm the reason why our study showed this greater preponderance of female respondents, we included this as a potential limitation and presented examples of previous studies performed in samples of individuals with gluten-related disorders from other countries that presented these same discrepancies. A previous example of an Argentinian study that evaluated quality of life of celiac patients and showed an even higher proportion of female respondents was also presented. Finally, studies that prove that female individuals tend to be more concerned about their health condition and therefore participate more in surveys were also cited.
Lines 307-312 of the previously received manuscript:
“A possible limitation of this study may be that our sample had a higher proportion of female respondents (88.7%). However, other studies with gluten-related disorders from other countries also exhibited this trend, with over 70% of female respondents [2,14,33,36]. Previous studies on HRQoL of celiac patients performed in the Argentinian population showed as high as 90% of female respondents 21. Moreover, females tend to generally be more concerned about their health and, therefore, participate more in health surveys than males [14,46]”
Q4: Discussion
Line 291: The phrase “The most recent update of the CD law is repeated”, please eliminate one of them
A4: Duplicate phrase has been deleted.
Q5: Conclusions
You can remark that the study was not presenctial and this way of action may have influenced also n the results obtained
A5: We assessed this as a potential limitation. However, because of the broad internet coverage and free internet access reported in Argentina (lines 313-315: “Another potential limitation could be the selection bias due to the online distribution of the survey. However, according to a recent international survey, 93.1% of the Argentinians have internet access, and many cities provide free internet access in public places 47,48.”) we can´t conclude this affected our final n.
Q6: Instead of asking about the influence of various factors in the last 2 weeks would be better to as about the last month
A6: All questions asked were based on the respondent’s experience in the past two weeks to reduce memory bias. A time frame of two weeks is indicated in the assessment of some questionnaires developed to evaluate health-related quality of life in population (WHOQOL; WHOQOL-BREF), developed by the World Health Organization. Also, we followed the original questionnaire previously published by Hauser et al. We recognize that different time frames may be necessary for particular uses of the instruments, mainly in chronic conditions. However, since we opt to follow the original instrument, we used the same time frame.
References are good and well selected
Reviewer 2 Report
Overall the work is well structured and organized.
Minor revision:
1) We ask to improve the English language;
2) Incorrect symbols appear in the questionnaire;
3) The bibliography should be updated;
Author Response
Thank you for your comments, questions and for taking your time to review our study.
Minor revision:
- We ask to improve the English language;
A: An English native speaker revised the manuscript.
- Incorrect symbols appear in the questionnaire;
A: We revised the questionnaire. We apologize for our typos; they were addressed in the manuscript. Thank you for your observation.
- The bibliography should be updated;
A: It was done.
Reviewer 3 Report
The authors carried out a useful study to explore quality of life of celiac patients from Argentina. However I have detecter several major points to be clarified.
Major points
Abstract starts declaring that the '... study aimed to translate, culturally adapt, validate, and apply a Celiac Disease Quality of Life', however within the abstract there are no results on the translation process and culturally adaptation, but only there are results on the application of the survey tool.
Methods: Translation of the questionnaire was performed from English version to Argentinian Spanish. The original questionnaire was validated in German, therefore, the authors should have translated from German to Argentinian Spanish.
Survey tools to assess public health problems should be systematically tested in order to ensure the reliability of data derived from their use. Specifically, in the context of gluten-related disorders, some of this type of tools have been developed and they have allowed to expand the knowledge on the characteristics of the population suffering these conditions.
In order to use these tools in a speaking-population other than the original speaking-population target, a simple translation is not enough. Thus, authors should show results on the analysis of translation/back-translation, in other words what type of analysis did the authors use to evaluate discrepancies and consensus between translations/back-translations?.
I suggest to include a figure involving a graphical summary of the methods. Results There are no results on the analysis of translation/back-translation. Please include them.
The table 2 is very number heavy. I suggest to design an attractive figure (it could be even a figure with several parts) involving data from this table. The authors should to include the response rate during the survey application.
Discussion: Casellas et al., (2013) (Rev esp enfeRm dig (Madrid) 2013, 105(10): 585-593) had previously translated to Spanish the same questionnaire by Häuser et al. (2007). However, the authors did not discuss differences or coincidences between the own results and those by Casellas et al.
The discussion on the results from the application of the survey should to acknowledge as a limitation the fact of internet platforms use reduces the response rate. Although the authors correctly acknowledged the need to increase the number of respondents as a limitation of the study, they should to include information from Aramburo-Galvez's studies reporting that surveys using internet platforms to collect data had a response rate close to 9% (Arámburo-Gálvez, 2019. Translation, cultural adaptation, and evaluation of a brazilian portuguese questionnaire to estimate the self-reported prevalence of gluten-related disorders and adherence to gluten-free diet. Medicina, 55(9), 593).
Instead, when survey was face to face performed, the response rate increased up to 90% (Arámburo-galvez et al., 2020 Prevalence of Adverse Reactions to Gluten and People Going on a Gluten-Free Diet: A Survey Study Conducted in Brazil. Medicina, 56(4), 163).
Minor points: In the discussion section (page 4, lines 279-280) authors declare 'Although it was not our objective to evaluate the adherence to GFD, we found a surprisingly high proportion of respondents answered they were “always” on a GFD, 83.9%'.
But I consider that is not surprising since the study by Cabrera-Chávez et al., (2017) (Cabrera-Chávez et al., 2017.Prevalence of Self-Reported Gluten Sensitivity and Adherence to a Gluten-Free Diet in Argentinian Adult Population. Nutrients 9(1): 81.) showed that 0.91% of Argentinian population follows the GFD and from the same study it was estimated the celiac disease prevalence in 0.58%. Thus, I suggest to include a comparison to this research.
In several parts of the manuscript the authors declare the semantic evaluation of the Argentinian Spanish version, however I have detected a couple of points where semantic should be improved: page 10, line 629 '...demostraron no ser comprensivos o no entender al respecto de tu celiaquía?'; page 11, line 646: '... demostraron no ser comprensivos o no entender al respecto de tu celiaquía?' I consider that it should be '... demostraron no ser comprensivos o no entender tu situación/condición como paciente celiaco?' Page 9, line 33. 'recreative' should be 'recreativo' Page 10, line 586. 'excluido' should be 'excluído'
Author Response
Thank you for your comments, questions and for taking your time to review our study.
The authors carried out a useful study to explore quality of life of celiac patients from Argentina. However I have detecter several major points to be clarified.
Major points
1. Abstract starts declaring that the '... study aimed to translate, culturally adapt, validate, and apply a Celiac Disease Quality of Life', however within the abstract there are no results on the translation process and culturally adaptation, but only there are results on the application of the survey tool.
Reply: Results if these phases have been added (current lines 31- 40).
2. Methods: Translation of the questionnaire was performed from English version to Argentinian Spanish. The original questionnaire was validated in German, therefore, the authors should have translated from German to Argentinian Spanish.
Reply: In page 159 from Häuser et al, 2007 the author says: “The German version of the CDQ was adapted to the English language by a forward-backward translation by 2 professional bilingual translators (1 medical, 1 nonmedical) and 2 German medical doctors. The English version of the CDQ is included in the appendix.”
Although the questionnaire was validated in the German population, the English version of the CDQ was the one published as appendix from the original article making the English version the original available one to be consulted for the scientific community. This English version was already used in previous studies that translated and culturally adapted this same questionnaire (Aksan et at., 2015 and Pratesi et al., 2018)
3. Survey tools to assess public health problems should be systematically tested in order to ensure the reliability of data derived from their use. Specifically, in the context of gluten-related disorders, some of this type of tools have been developed and they have allowed to expand the knowledge on the characteristics of the population suffering these conditions.
In order to use these tools in a speaking-population other than the original speaking-population target, a simple translation is not enough. Thus, authors should show results on the analysis of translation/back-translation, in other words what type of analysis did the authors use to evaluate discrepancies and consensus between translations/back-translations?
Reply: As explained in item 2. “Materials and Methods” this study was performed in four stages, each of them detailed in a separate item of the methodology and performed following validated and extensively used validation procedures in the area of public health as Guillemin guidelines and Delphi method. Consensus between judges/experts involved in phase 2 (Cultural adaptation and semantic evaluation) was objectively evaluated by applying Likert scale to the available answer options set to evaluate clarity, content validation, and semantic validation of each of the propossed questions.
4. I suggest to include a figure involving a graphical summary of the methods.
Reply: Figure has been included (Figure 1).
5. Results: There are no results on the analysis of translation/back-translation. Please include them.
Reply: Results have been included in item 3.1 from Results.
6. The table 2 is very number heavy. I suggest to design an attractive figure (it could be even a figure with several parts) involving data from this table. The authors should to include the response rate during the survey application.
Reply: Although we agree that figures are more visually attractive that tables, they must also be informative and display information in a comparable manner for them to be useful. We don´t think our data will fit an image while maintaining those characteristics.
7. Discussion: Casellas et al., (2013) (Rev esp enfeRm dig (Madrid) 2013, 105(10): 585-593) had previously translated to Spanish the same questionnaire by Häuser et al. (2007). However, the authors did not discuss differences or coincidences between the own results and those by Casellas et al.
Reply: Although we cited this previously published scientific article (reference 11 from our manuscript) the aim of our study was not to compare our results with the ones from other populations. This would have required further statistical analyses that were not intended to be performed on the current study. We are aware that both countries’ official language is Spanish however, the type of Spanish spoken is very different. Moreover, Argentinian celiac disease patients´ culture, access to gluten-free food, public health policies, among others, are very different from that of European countries. We considered and shortly discussed some general findings that called our attention when comparing the results obtained in the Brazilian survey with ours since we believe the before mentioned population characteristics are keener to be similar between South American countries. Brazil is the only other South American country that, to our knowledge, has applied this survey tool.
8. The discussion on the results from the application of the survey should to acknowledge as a limitation the fact of internet platforms use reduces the response rate. Although the authors correctly acknowledged the need to increase the number of respondents as a limitation of the study, they should to include information from Aramburo-Galvez's studies reporting that surveys using internet platforms to collect data had a response rate close to 9% (Arámburo-Gálvez, 2019. Translation, cultural adaptation, and evaluation of a brazilian portuguese questionnaire to estimate the self-reported prevalence of gluten-related disorders and adherence to gluten-free diet. Medicina, 55(9), 593).
Instead, when survey was face to face performed, the response rate increased up to 90% (Arámburo-galvez et al., 2020 Prevalence of Adverse Reactions to Gluten and People Going on a Gluten-Free Diet: A Survey Study Conducted in Brazil. Medicina, 56(4), 163).
Reply: As stated before, these were acknowledged as a probable limitation. Other important factors were also mentioned, like the availability of free internet connection in public places coved by a national program of connectivity. Reason why we did mention the probable limitations but did not address it necessarily as an issue. We have also discussed this topic with the judges (all Argentinian individuals) that were involved in previous stages of the survey tool development and members of celiac associations, they all agreed this way of distributing the questionnaire will be the one that would achieve greater reach.
9. Minor points: In the discussion section (page 4, lines 279-280) authors declare 'Although it was not our objective to evaluate the adherence to GFD, we found a surprisingly high proportion of respondents answered they were “always” on a GFD, 83.9%'.
But I consider that is not surprising since the study by Cabrera-Chávez et al., (2017) (Cabrera-Chávez et al., 2017.Prevalence of Self-Reported Gluten Sensitivity and Adherence to a Gluten-Free Diet in Argentinian Adult Population. Nutrients 9(1): 81.) showed that 0.91% of Argentinian population follows the GFD and from the same study it was estimated the celiac disease prevalence in 0.58%. Thus, I suggest to include a comparison to this research.
Reply: We excluded the word surprisingly. The study performed in the comment above was performed in the general population and only 7 self-reported celiac participants provided information about their gluten-free diet adherence status. These numbers as well as ours as still very low to make appropriate comparisons without the corresponding statistical analysis. However, we inserted a paragraph mentioning the study and the limitations of data comparison.
10. In several parts of the manuscript the authors declare the semantic evaluation of the Argentinian Spanish version, however I have detected a couple of points where semantic should be improved: page 10, line 629 '...demostraron no ser comprensivos o no entender al respecto de tu celiaquía?'; page 11, line 646: '... demostraron no ser comprensivos o no entender al respecto de tu celiaquía?' I consider that it should be '... demostraron no ser comprensivos o no entender tu situación/condición como paciente celiaco?'
Reply: We appreciate your observation and suggestion. This is a modification we can´t make at this point because the questionnaire was already submitted and applied to IRB review the way it is shown in the paper. The originally submitted questionnaire is the one we made available for publication. However, we will consider this suggestion for further implementations of the survey tool.
Page 9, line 33. 'recreative' should be 'recreativo' Page 10, line 586. 'excluido' should be 'excluído' A: The errors mentioned were typos, they were addressed in the manuscript. Thank you for your observation.
Round 2
Reviewer 3 Report
The authors have addressed some of the major points of my previous review, however they did not attend the changes suggested and did not enough support why they did so in two major points:
In major point 7 the authors replied "... the aim of our study was not to compare our results with the ones from other populations. This would have required further statistical analyses that were not intended to be performed on the current study. We are aware that both countries’ official language is Spanish however, the type of Spanish spoken is very different.", then, it is difficult to me understand why the authors consider that comparisons among results from published data are not relevant, moreover, the suggested discussion was related to comparison among characteristics of the validated survey in Spanish (clarity, comprehensibility, repeatability, etc.). I consider that discussion in scientific papers should take into account the current knowledge published by other authors and not only the results by the same group of work.
Also the authors replied "...Brazil is the only other South American country that, to our knowledge, has applied this survey tool.". I agree, but a similar survey was applied in Mexican celiac population (Ramírez-Cervantes et al., 2015. Characteristics and factors related to quality of life in Mexican Mestizo patients with celiac disease. BMC gastroenterology, 15(1), 1-7) and the authors should compare results (for instance, how is the quality of life among Mexican celiac patients compared to Argentinian Population).
In point 7 the authors replied "As stated before, these were acknowledged as a probable limitation." I consider it is not a possible limitation but it is a true limitation since several authors have reported the lower response rate when internet platforms are used instead of face to face interviews. Againly, I suggest to include (in discussion section) information from Aramburo-Galvez's studies reporting that surveys using internet platforms to collect data had a response rate close to 9% (Arámburo-Gálvez, 2019. Medicina, 55(9), 593) but when the survey is applied face to face the response rate increased up to 90% (Arámburo-galvez et al., 2020. Medicina, 56(4), 163).
Minor point:
If the semantic modification of the survey is not possible, please avoid the mentions to semantic evaluation throughout the manuscript.
The authors did not attend some of the changes previously suggested. The major point is the neglecting by the authors to consider the background on the topic of celiac disease carried out in Latin America
Author Response
Q1: In major point 7 the authors replied "... the aim of our study was not to compare our results with the ones from other populations. This would have required further statistical analyses that were not intended to be performed on the current study. We are aware that both countries’ official language is Spanish however, the type of Spanish spoken is very different.", then, it is difficult to me understand why the authors consider that comparisons among results from published data are not relevant, moreover, the suggested discussion was related to comparison among characteristics of the validated survey in Spanish (clarity, comprehensibility, repeatability, etc.). I consider that discussion in scientific papers should take into account the current knowledge published by other authors and not only the results by the same group of work.
A1: The Spanish study was not performed using the same survey tool. The reference and citation was added by mistake and was taken out from the manuscript.
Q2: Also the authors replied "...Brazil is the only other South American country that, to our knowledge, has applied this survey tool.". I agree, but a similar survey was applied in Mexican celiac population (Ramírez-Cervantes et al., 2015. Characteristics and factors related to quality of life in Mexican Mestizo patients with celiac disease. BMC gastroenterology, 15(1), 1-7) and the authors should compare results (for instance, how is the quality of life among Mexican celiac patients compared to Argentinian Population).
A2: We have stated within the manuscript that our data does not have statistical power to make comparisons between ours and other populations. This has been mentioned, in other words, when assessing the limitation of having a small n, a point that this same reviewer already acknowledged in his/her first round of comments. We believe the importance of assessing the weaknesses of the study is not only to follow procedure but to be aware of them and act accordingly for the processing of the data. We do not feel comfortable doing comparisons with the n we have before performing the appropriate statistical treatment of the data to make samples of our and others studies equivalent. The intention of this work (as also mentioned before) was to validate the survey and see if this will be a useful way of assessing HRQoL in Argentinian celiac patients. It is our intention to, after proving its suitability, apply it to a larger n and then compare the findings to that of other studies.
Q3: In point 7 the authors replied "As stated before, these were acknowledged as a probable limitation." I consider it is not a possible limitation but it is a true limitation since several authors have reported the lower response rate when internet platforms are used instead of face to face interviews. Againly, I suggest to include (in discussion section) information from Aramburo-Galvez's studies reporting that surveys using internet platforms to collect data had a response rate close to 9% (Arámburo-Gálvez, 2019. Medicina, 55(9), 593) but when the survey is applied face to face the response rate increased up to 90% (Arámburo-galvez et al., 2020. Medicina, 56(4), 163).
A3: The selection bias due to the online distribution of the survey has been addressed at the end of the discussion.
Minor point:
Q4: If the semantic modification of the survey is not possible, please avoid the mentions to semantic evaluation throughout the manuscript.
A4: The semantic modification was possible and was performed during the study. What we presented to the editor and reviewers was the final tool originated from all steps listed in the methodology we cannot take any already performed step from the methodology and the present results that considered this step as well as we cannot modify the survey tool after having applied it (meaning sent to the celiac patients) as the result were generated using that specific tool. Although we said that we appreciate the recommendation and that we will be happy to consider it for a future survey, making the suggested modification at this point (after the ethics committee approval and application) would be equivalent to falsifying data as we would be presenting a survey tool different from the one that we actually used.
Q5 The authors did not attend some of the changes previously suggested. The major point is the neglecting by the authors to consider the background on the topic of celiac disease carried out in Latin America.
A5 - We have now addressed all the issues addressed by the reviewer.
Round 3
Reviewer 3 Report
Authors have ignored my suggestions and only have eliminated the former reference (11) by Casellas et al., 2013 saying that this reference was a mistake and "The Spanish study was not performed using the same survey tool. The reference and citation was added by mistake and was taken out from the manuscript" but this is not true. Casellas et al. (2013), declared that they used the "Coeliac Quality of Life Survey" by Häuser et al., 2007 (Häuser W, Gold J, Stallmach A, Caspary WF, Stein J. Development and validation of the Celiac Disease Questionnaire (CDQ), a disease-spe- cific health-related quality of life measure for adult patients with celiac disease. J Clin Gastroenterol 2007;41:157 66). I Have reviewed the cited articles by the authors (current references 17,19,18, and 14), and this (Häuser et al., 2007) is the same reference that the authors cite as the original source of the questionnaire they used and also have been used in "France [17], Turkey [19], Italy [18], , and Brazil [14]." (introduction section page 2, lines 82-83). Then, if the original inclusion of the reference Casellas et al., (2013) was a mistake, consequently the authors should have eliminated also the self-citations (19), (18), and (14) and even the reference (17).
The authors still declaring that "Unlike Brazil, the only other South American country where this questionnaire was applied..." ignoring my previous suggestion to include the study by Real-Delor and Centurion-Medina (2018) that used the same questionnaire in a South American population from Paraguay (Open Acces Article: Real-Delor & Centurion-Medina, 2018. Quality of Life in Paraguayan Adults with Celiac Disease. DUAZARY, 15(1), 61-70).
Author Response
Dear reviewer,
We read, thoroughly assessed, and responded to all your previous comments. Please note that the comment you sent as related to the Paraguayan study is tight to the study performed by Casellas et al (2013) as they cited this study (Casellas et al) when referring to the type of survey tool they chose to perform the research. The chosen survey tool was the CD-QOL (Dorn et at, 2010) and not the CDQ (Häuser et al, 2007). Please see the fragment of Real-Delor and Centurión-Medina (2018) - Paraguayan study- where they state the before mentioned´-page 63:
“Para esta investigación se aplicó el cuestionario CD-QOL validado al idioma castellano15,17. Este instrumento consiste e 20 preguntas cuyas respuestas se puntúa con una escala de Likert de 1 al 5. La sumatoria final varía desde 0 puntos (peor calidad de vida) hasta 100 (mejor calidad de vida). Las preguntas se refieren a cuatro dimensiones: disforia, limitaciones, salud y tratamiento inadecuado15,17.”
Cited references:
- Rodríguez Almagro J, Hernández Martínez A, Lucendo AJ, Casellas F, Solano Ruiz MC, Siles González J. Health-related quality of life and determinant factors in celiac disease. A population-based analysis of adult patients in Spain. Rev Esp Enferm Dig. 2016 Apr;108(4):181-9.
- Casellas F, Rodrigo L, Molina-Infante J, Vivas S, Lucendo AJ, Rosinach M, et al. Transucultural adaptation and validation of the Celiac Disease Quality of Life (CD-QOL) Survey, a specific questionnaire to measure quality of life in patients with celiac disease. Rev Esp Enferm Dig. 2013 Nov-Dec;105(10):585-93.
Also please find the manuscript fragments from Rodríguez Almagro et at (2016) and Casellas et al (2013)- the previously cited papers- below, where they specifically state that they chose to use the CDQOL (originally developed by Dorn et al, 2010) rather than the CDQ (Häuser et al, 2007). Casellas et al also explained why they decided NOT to use the CDQ (Häuser et al, 2007).
Rodríguez Almagro et al (2016) -citation number 15 from Real-Delor and Centurión-Medina (2018):
“HRQL was assessed with the aid of a specific CD-QOL questionnaire, translated into Spanish and validated for use in a Spanish setting (14) with adult patients. The CD-QOL consists of 20 items examining four dimensions: dysphoria, limitations, health concern, and inadequate treatment. The questions are answered by the patient, who scores them on a Likert scale from 1 (total disagreement) to 5 (total agreement). The CD-QOL questionnaire produces an overall rating between 0 (worst quality of life) and 100 (best quality of life) points. Each of the four dimensions is expressed with the same scale of 0-100.
Cited reference:
- Casellas F, Rodrigo L, Molina-Infante J, et al. Transcultural adaptation and validation of the Celiac Disease Quality of Life (CD-QOL) Survey, a specific questionnaire to measure quality of life in patients with celiac disease. Rev Esp Enferm Dig 2013;105:585-93. DOI: 10.4321/S1130-01082013001000003.
Casellas et al (2013)-page 586 -citation number 17 from Real-Delor and Centurión-Medina (2018):
“Four specific questionnaires to measure HRQOL in English-speaking celiac patients are available. Two of these, the Celiac Disease DUX (CDDUX) and the Celiac Disease Quality of Life Instrument for North American Children, were designed for children (11,12), while the other two instruments, the Coeliac Quality of Life Survey (CD-QOL) and the Celiac Disease Questionnaire (CDQ), were designed for adult celiac patients (13,14). However, none of them have been translated and validated in the Spanish-speaking population.”
“Due to the lack of HRQOL measurement questionnaires specific for celiac disease in Spanish, all HRQOL studies conducted have used generic questionnaires, having less sensitivity and capacity to detect changes over time. The purpose of this study was to translate into Spanish the specific questionnaire for celiac disease in adults of the Coeliac Quality of Life Survey (CD QOL), and to establish whether the Spanish version retains adequate psychometric properties. The CD-QOL is a self-administered questionnaire consisting of 20 questions distributed into four dimensions –dysphoria, limitations, health concerns, and inadequate treatment– that should be answered in a Likert scale. This questionnaire was chosen because it was designed and validated for the adult population with celiac disease, has been shown to be equally valid in other European Union languages (23) and, unlike CDQ, is less focused on physical and psychical symptoms and more focused on disease-related needs, according to authors of the original CD-QOL.
Cited reference:
- Häuser W, Gold J, Stallmach A, Caspary WF, Stein J. Development and validation of the Celiac Disease Questionnaire (CDQ), a disease-specific health-related quality of life measure for adult patients with celiac disease. J Clin Gastroenterol 2007;41:157-66.
- Dorn DS, Hernández L, Minaya MT, Morris CB, Hu Y, Leserman J, et al. The development and validation of a new coeliac disease quality of life survey (CD-QOL). Aliment Phar Ther 2010;31:666-75.
- Zingone F, Iavarone A, Tortora R, Imperatore N, Pellegrini L, Russo T, et al. The Italian translation of the celiac disease-specific quality of life scale in celiac patients on gluten free diet. Dig Liver Dis 2013;45:115-8.
This last reference (23) is from a study that also used the CDQOL. For confirmation please see below the specific fragment of the manuscript where they state this – page 115 “Materials and methods”:
“The Italian translation of the CD-QOL questionnaire, devised and previously validated in English by Dorn et al. [8] was performed according to the Rome translation project and approved by the Rome Foundation appointed clinician”
Cited reference:
[8] Dorn SD, Hernandez L, Minaya MT, et al. The development and validation of a new coeliac disease quality of life survey (CD-QOL). Alimentary Pharmacology and Therapeutics 2010;31:666–75.
The evidence shown above supports our statement "...Brazil is the only other South American country that, to our knowledge, has applied this survey tool." In addition, please find below a simplified chart we have prepared to highlight some important differences between questionnaires (survey tools) CDQOL and CDQ. Although they were designed to ultimately evaluate the health-related quality of life of celiac patients, they do not assess the same dimensions of health and therefore the approach they use is different (all the information included in the below chart can be easily found in the previously cited fragments -CDQOL related- and in our manuscript -CDQ related- also please feel free to cross-check what was referenced with the information found in the original manuscripts from the development of both questionnaires).
|
|
CDQOL (Dorn et al, 2010) |
CDQ (Häuser et al, 2007) |
|
Number of questions |
20 |
28 |
|
Range of applied Likert scale for each question |
1-5 |
1-7 |
|
Assessed health dimensions |
dysphoria, limitations, health concern, and inadequate treatment |
emotions, social, worries, and gastrointestinal |
|
Original reference from the study that created the survey tool (questionnaire) |
Dorn DS, Hernández L, Minaya MT, Morris CB, Hu Y, Leserman J, et al. The development and validation of a new coeliac disease quality of life survey (CD-QOL). Aliment Phar Ther 2010;31:666-75. |
Hauser W, Gold J, Stallmach A, Caspary WF, Stein J (2007) Development and validation of the celiac disease questionnaire (CDQ), a disease specific health-related quality of life measure for adult patients with celiac disease. J Clin Gastroenterol 41: 157–166 |
We included all full references cited for the reviewer, editor and any other appropriate editorial member to be able to cross-check all the referenced manuscripts’ fragments. Please feel free to do so.